

# Universal finite-size amplitude and anomalous entanglement entropy of $z = 2$ quantum Lifshitz criticalities in topological chains

Ke Wang$^\star$ and Tigran A. Sedrakyan

Department of Physics, University of Massachusetts, Amherst, MA 01003, USA

$\star$ kewang@umass.edu

## Abstract

We consider Lifshitz criticalities with dynamical exponent $z = 2$ that emerge in a class of topological chains. There, such a criticality plays a fundamental role in describing transitions between symmetry-enriched conformal field theories (CFTs). We report that, at such critical points in one spatial dimension, the finite-size correction to the energy scales with system size, $L$, as $\sim L^{-2}$, with universal and anomalously large coefficient. The behavior originates from the specific dispersion around the Fermi surface, $\epsilon \propto \pm k^2$. We also show that the entanglement entropy exhibits at the criticality a non-logarithmic dependence on $l/L$, where $l$ is the length of the sub-system. In the limit of $l \ll L$, the maximally-entangled ground state has the entropy, $S(l/L) = S_0 + 2n(l/L)\log(l/L)$. Here $S_0$ is some non-universal entropy originating from short-range correlations, and n is a half-integer or integer depending on the degrees of freedom in the model. We show that the novel entanglement originates from the long-range correlation mediated by a zero mode in the low energy sector. The work paves the way to study finite-size effects and entanglement entropy around Lifshitz criticalities and offers an insight into transitions between symmetry-enriched criticalities.



# 1  Introduction

A class of criticalities separate gapped symmetry protected phases [1–3] (SPTs) and topologically trivial ones. At these criticalities usually the system disperses linearly, $\epsilon = \pm v_F k$, around the Fermi surface, and the low-energy effective physics is described by conformal field theories [4–6] (CFTs). Several universal features characterize conformal critical points. One notable feature for quantum one-dimensional (1D) systems is the universal finite-size amplitude [7] together with the emergence of the universal characteristic of CFTs, the central charge, $c$. Namely, the finite-size correction to the ground state energy $E(L)$, e.g., in case of open boundary condition (b.c), always contains a universal term $c\pi/24L$. The other universal feature is the logarithmic entanglement entropy [8], e.g. , $S \sim c \ln(L)/6$ in the case of periodic b.c.

Topologically distinct and gapped phases are reached by adding the mass to the CFT criticalities [9]. A simple example is the hamiltonian $h(i\partial_x) = v_F \sigma_y i\partial_x + m\sigma_x$ and sign$(m)$ is an integer to distinguish phases. Here $\sigma_{x,y}$ are Pauli matrices. Universal features also appear around the topological phase transitions [10], e.g., the finite-size correction emerges as a universal function of scale, $\omega = mL$.

Recently, it has been observed that CFT critical phases can have non-trivial topology and host boundary modes. Such criticalities are dubbed symmetry-enriched criticalities [11–13] or called gapless SPTs [14,15]. At the transition between two symmetry-enriched CFTs, non-CFT criticalities can emerge [16]. The simplest case is the Lifshitz criticality [17–23] with dynamical exponent $z = 2$. Its role as a criticality between gapless SPTs is similar to CFT critical points separating gapped SPTs. Namely, one can reach topologically distinct gapless phases by adding velocity term $v$ to $z = 2$ critical point. A simple Hamiltonian illustrates this fact,

$$h(-i\partial_x) = v\sigma_y(i\partial_x) + u\sigma_x\partial_x^2. \tag{1}$$

Here $v$ is the velocity, and $u$ is the curvature of the spectrum. The case with $v = 0$ corresponds to a non-CFT criticality, referred to as $\Pi$ throughout this paper. With appropriate boundary conditions, one can find the eigenstate, $\psi(x)$, of the Hamiltonian Eq. 1, exhibiting boundary modes at sign$(v) > 0$ which however disappear at sign$(v) < 0$. Thus, adding velocity perturbations to the $z = 2$ criticality generates two gapless phases: one topologically trivial and another non-trivial. This finding is supported by the detailed calculation in the Sec. 5.

In spite of its fundamental role of describing transitions between symmetry-enriched CFTs, the understanding of universal features of $z = 2$ critical points (with the dispersion $\epsilon \sim \pm k^2$) is still lacking. In this letter, we aim to understand the universal properties of $\Pi$ criticality from two aspects: the study of the energy and entanglement entropy of the ground state. To this end, we consider two concrete lattice models and develop the low energy field theoretical description of the criticality. Lattice models considered below are Majorana/Kitaev chains [16, 24] with next-nearest neighbor terms from BDI symmetry class [24–27] and the generalized Su-Schrieffer–Heeger (SSH) model [28,29] with next-next-nearest neighbor terms belonging to the AIII symmetry class.

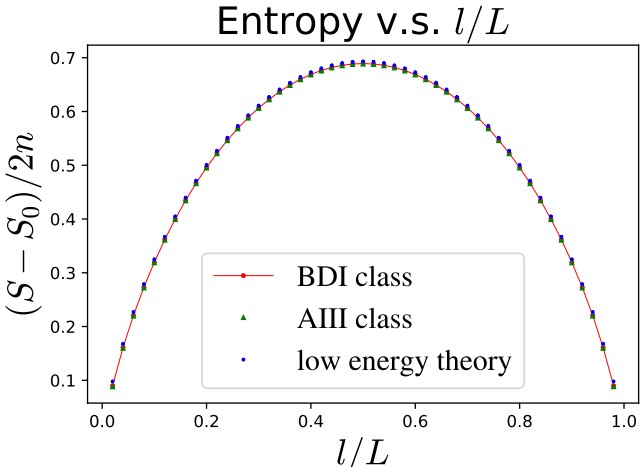

Figure 1: (Color online) Entanglement entropy $(S - S_0)/2n$ is plotted versus $l/L$. Here $S_0$ is the non-universal constant entropy, $l$ is the size of subsystem, $n = 1/2$ for the Majorana chain in BDI class and $n = 1$ for the SSH model in AIII class. Three sets of data, including entropy of the Majorana chain, SSH model, and low energy theory, all fall into the same *universal* curve. The function, representing the plotted curve, is exactly the $l/L$-dependent term in Eq. 3.

The first result of the present letter corresponds to the ground state energy $E(L)$ as a function of the system size, $L$. At open boundary condition, the finite-size corrections [30–32] to $E(L)$ exhibit a universal behavior and read

$$E(L) = L\epsilon + b - nu\frac{J}{L^2} + O(L^{-3}). \tag{2}$$

Here $\epsilon$ is average bulk energy, $b$ is the boundary energy, and $n \in \mathbb{Z}^+/2$ depends on degrees of freedom of the underlying field theory: $n = 1/2$ for the Majorana chain and $n = 1$ for the SSH model under consideration. The amplitude $J$ is $J \simeq 0.887984$, which is universal for two lattice models and the low-energy field theory giving the same value. This indicates a possible set of rich phenomena of finite-size scaling functions around this criticality [10, 33–35]. We have checked that velocity perturbations modify $A$ into a universal scaling function of $\omega = Lv$, and the function is sensitive to the topological nature of CFTs.

We also find that the entanglement entropy [8] exhibits an interesting dependence on $l/L$. At periodic b.c, the von Neumann entropy of the maximally-entangled ground state is given by

$$S \simeq S_0 + 2n \cdot \left[\frac{l}{L}\ln\left(\frac{L}{l} - 1\right) - \ln\left(1 - \frac{l}{L}\right)\right]. \tag{3}$$

Here $l$ is the length of the subsystem, and $S_0$ is a non-universal constant. At the limit $l/L \ll 1$, $S$ has a simple asymptote $\sim (l/L)\log(l/L)$, which is non-logarithmic. The $l/L$-dependent term is found to be universal, plotted in Fig. 1. Below we start with a definition of lattice models and observe the emergence of $\Pi$ criticality.

The remainder of the paper is organized as follows. In Section II, we start with a definition of lattice models and observe the emergence of $\Pi$ criticality. In Section III, the analytical study of the universal finite-size amplitude at the criticality is presented. Section IV presents the derivation of the anomalous entanglement entropy corresponding to the $\Pi$ criticality. We also extend our results to the non-zero velocity case and discuss the quantization condition. Finally,

the conclusions are presented in Section VI. We refer the reader to Appendices A and B for the details of the calculations.

## 2 Lattice models and Criticality

We consider two concrete one-dimensional lattice models supporting the Lifshitz $\Pi$ criticality. One is the Majorana chain, containing both nearest site and next-nearest site hoppings and pairings. The Hamiltonian is given by

$$H_{\text{Majorana}} = \sum_n t_0 \tilde{\gamma}_n \gamma_n + t_1 \tilde{\gamma}_n \gamma_{n+1} + t_2 \tilde{\gamma}_n \gamma_{n+2}. \tag{4}$$

Here $\{\gamma_n, \tilde{\gamma}_n\}$ are two Majorana fermions at the same physical site, and constants $t_i \in \mathbb{R}$, with $i = 0, 1, 2$. The model is schematically shown in Fig. 2a. Note that the model belongs to the BDI class of Cartan's classification of symmetric spaces. A critical line of the model, where the gap closes, corresponds to the case $t_2 + t_0 = t_1$. One can observe three distinct critical behaviors in this situation: (1) when $0 < t_2/t_1 < 1/2$, the low-energy sector is described by Majorana CFT and two localized Majorana modes. (2) When $1 > t_2/t_1 > 1/2$, the low-energy description is a single Majorana CFT. (3) At $t_2 = t_0 = t_1/2$, the $\Pi$ criticality emerges around $k = \pi$ in the Brillouin zone. The Hamiltonian around the Fermi surface, in Bogoliubov-de-Gennes (BdG) formalism, can be written as

$$H_{\text{FS}} = u \int dx \Psi^{\dagger}(x) \sigma_x \partial_x^2 \Psi(x). \tag{5}$$

Here $\Psi(x) = (\psi(x), \psi^{\dagger}(x))^T$ and $\psi(x)$ is the spinless fermion operator in the continuous space.

The second model under consideration is the generalized SSH model belonging to the AIII symmetry class. The model is schematically shown in Fig. 2b. The Hamiltonian includes nearest-neighbor and next-next-nearest neighbor hoppings of fermions $c^{(\dagger)}$ and is given by

$$H_{\text{ssh}} = \sum_n u_0 c_{n,A}^{\dagger} c_{n,B} + \sum_{i=1,2} u_i c_{n,B}^{\dagger} c_{n+i,A} + h.c.. \tag{6}$$

The model is defined on a bipartite lattice with $A$ and $B$ sublattices and real hopping parameters. It has a similar phase diagram as the model of Majorana chains identified above. Here the criticality $\Pi$ emerges around $k = \pi$ when $u_0 = u_2 = u_1/2$. Now the Hamiltonian around the Fermi surface is described by Eq. 5 but with $\Psi(x) = (\psi_A(x), \psi_B(x))$.

Below, we will identify a set of universal properties inherent for theories at $\Pi$ criticality and present the low-energy field theory description of the system yielding these universal properties.

## 3 Universal finite-size amplitude

The present section studies the finite-size correction to the ground state energy, $E(L)$, of the system at open boundary conditions at criticality $\Pi$.

In the lattice models under consideration, the computation of the finite-size amplitude of the ground state is similar to the method used in references [10, 33], that was applied to CFT

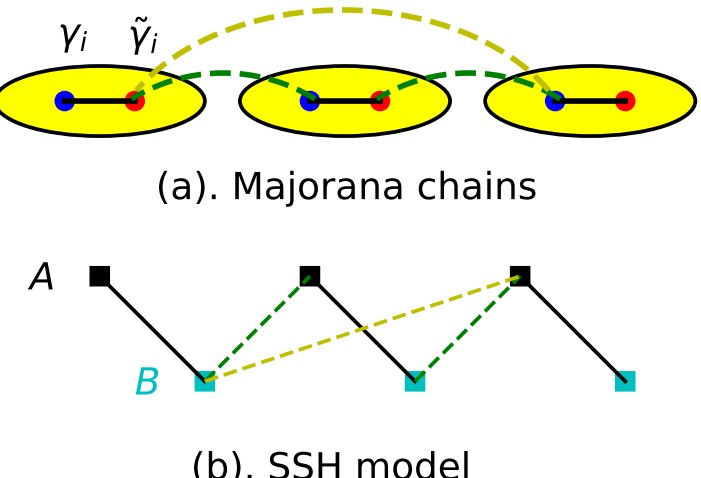

Figure 2: (Color online) Models with three unit cells are plotted to illustrate the hoppings and pairings. (a) Majorana chain. A single fermion is decomposed into two Majorana fermions, shown as blue and red dots. Black lines represent $t_0$ and dashed green/yellow lines represents $t_1/t_2$ hoppings (and pairings) in Eq. 4. (b) SSH model. Black/cyan rectangular dots represent $A/B$ sublattices. Black, green and yellow lines represent $u_0$, $u_1$ and $u_2$ hoppings in Eq. 6.

criticalities. The method[1] has an error bar, $\sim L^{-1}$. Here we report the results for the SSH model and Majorana chain: we pick $L = 500$ and $A$ is found to be 0.88441 for the SSH model and 0.88440 for the Majorana chain. Compared to the value of A below Eq. 2, errors are at the expected order, $\sim 10^{-3}$. The amplitude $A$ is universal because it originates from the long-wavelength degrees of freedom around the Fermi surface. Below, we will validate this point by deriving the amplitude $A$ from the low energy theory.

Consider the Hamiltonian Eq. 1 at $\nu = 0$. One special property of the operator $\sigma_x \partial_x^2$ is that the free wave and the bound states can belong to the same subspace of it. Namely, the eigenstates are $\psi_k(x) = \exp(ikx) \cdot \chi_-$ and $\psi_{ik}(x) = \exp(-kx) \cdot \chi_+$, which lie in the same energy level $\epsilon_k = uk^2$. Here $\chi_\pm$ satisfy $\sigma_x \chi_\pm = \pm \chi_\pm$ and $k \in (-\pi, \pi)$.

Now assume $h(-i\partial_x)$ acts on coordinate dependent wavefunctions with $x \in (0, L)$ and we impose open boundary conditions on wavefunctions. Namely, one has $\psi_1(0) = \partial_x \psi_1(0) = 0$ corresponding to the "left" boundary, and $\psi_2(L) = \partial_x \psi_2(L) = 0$ corresponding to the "right" boundary. These conditions describe the continuity of the wavefunction around boundaries, when one imposes the infinite velocity outside the segment $(0, L)$. Note that the wavefunction with the energy $\epsilon_k$ can be generally written as $\varphi_k(x) = \sum_{s=\pm} a_s \psi_{sk}(x) + b_s \psi_{isk}(x)$. Upon searching for solutions $\varphi_k(x)$, which obey the open boundary conditions, one arrives at the quantization condition (QC) of the momentum,

$$\cos kL + 1/\cosh kL = 0, \quad 0 < k < \pi, \tag{7}$$

different from conventional QC of Ising CFTs (that being $\cos kL = 0$). When $kL \gg 1$ in Eq. 7, the difference between the abovementioned QCs is exponentially small. However, when $kL \sim 1$, the difference is not negligible anymore. This difference indicates that there could be non-trivial finite-size effects. Solutions to Eq. 7 are shown in Fig. 3.

---

[1]The description of the method: The estimate of the boundary energy at a large system size of $10L$ is $\tilde{b} \simeq E(10L) - 10L\epsilon$. The terms responsible for the finite-size correction can be estimated as $J \simeq L^2(E(L) - L\epsilon - \tilde{b})/(1 - 10^{-2})$. The $\sim L^{-3}$ term in $E(L)$ contributes the error to $J$, which is proportional to $L^{-1}$.

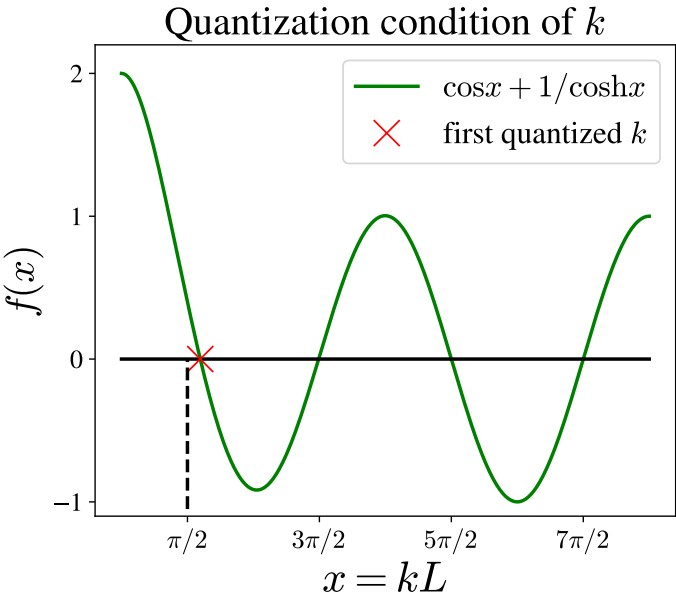

Figure 3: (Color online) Plot of the quantization condition in Eq. 7. The green curve plots the function $f(x) = \cos x + 1/\cosh x$ with $x = kL$. Intersections between $f(x)$ and $x$-axis determine quantized values of $k$. The first quantized value, located around $x_0 \simeq 1.875$, is marked by a red cross. This value deviate from the first quantized value in Ising CFT, $\pi/2$. Quanlitatively, the order of the amplitude $J$ can be aruged from this deviation: the deviation in the spectrum level is given by $x_0^2 - (\pi/2)^2 \sim 1$, which is the order of $J$.

To compute the ground state energy, one must perform summation over all the quasi-particle energies below the Fermi surface. Namely, $E(L) = -u \sum_{k \in \mathrm{QC}} k^2$. Note that all quantized solutions $k$ of Eq. 7 are invovled in the ground state energy. Summation in $E(L)$ can be written as a contour integral. Defining $z = kL$, $f(z) = \cos z + 1/\cosh z$ and taking the analytical continuation of $f(z)$, one finds

$$E(L) = -\frac{1}{2L^2} \oint_C \frac{dz}{2\pi i} z^2 \partial_z \ln f(z).$$  (8)

Here $C$ is the contour in complex plane $z = x + iy$, shown in Fig. 4. One can decompose $\ln f(z) = \ln \exp(iz) + \ln \exp(-iz)f(z)$. The first term of this decomposition plugged into Eq. (8) gives the bulk energy $L\epsilon$ of Eq. 2 while the second term gives the leading finite-size correction, $\propto J/L^2$. Further, one can deform the contour to obtain a regular integral over a single real variable. Namely, the $C$ is deformed to be contour $D$ at the cost of exponentially small error, shown in Fig. 4. We find,

$$J = -\mathrm{Re} \int_0^{+\infty} \frac{x^2 dx}{2\pi} \partial_x \ln \left( \frac{e^{(i-1)x} + 1}{2} + \frac{2}{e^x + e^{-ix}} \right).$$  (9)

The above integral is evaluated numerically, yielding $J = 0.887984$. We provide the detailed derivation and further evaluation of $J$ in the Appendix A. This analytically found constant matches the value of $J$ presented below Eq. 2. The value of $n$ can also be argued from the low-energy sector: $n = 1/2$ for BDI class is due to the property that the operator $\Psi(x) = (\psi(x), \psi^\dagger(x))^T$ is counted as $1/2$ degree of freedom, while $n = 1$ for AIII class is due to the fact that $\Psi(x) = (\psi_A(x), \psi_B(x))^T$ can be counted as $1$ degree of freedom. In this way, we proved the Eq. 2.

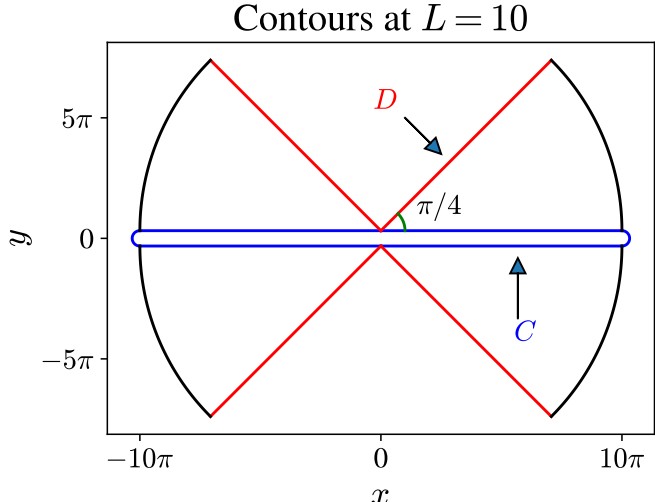

Figure 4: (Color online) Contours of integration. The blue contour $C$ corresponds to Eq. 8. One can deform the contour $C$ to $D$ (the red lines), since the integrand on the arc (the black line) is exponentially small and the function $\ln f(z)$ is holomorphic when $\text{Re}(z) \neq 0$ and $\text{Im}(z) \neq 0$.

## 4 Anomalous Entanglement entropy

The other universal data, which can be extracted from the Hamiltonian, is the entanglement entropy $S$. Below we take the Majorana chain as an example to illustrate the emergence of the anomalous entanglement. The consideration for the SSH model is similar.

At first glance, one may observe the eigenstates of Eq. 5 are not different from those of the gapped Hamiltonian ( where one replaces $k^2 \rightarrow m$ and $m \neq 0$ ). Thus one expects short-ranged correlations and constant (non-universal) entanglement entropy, known as features of a gapped 1D quantum system. However, the presence of zero-modes at the Fermi surface changes the situation. With periodic boundary conditions, the $k = 0$ eigenstate leads to the double degeneracy of the ground state. Tracing the maximally-entangled ground state, we find that the asymptotic correlation function is given by

$$\langle \eta_\alpha(x)\eta_\beta(y) \rangle \simeq \epsilon^{\alpha,\beta} \frac{2i}{L} e^{ik_F(x-y)}, \quad \text{when} \quad |x-y| \gg 1 \quad \text{and} \quad \alpha, \beta = 1, 2. \quad (10)$$

Here we relabeled the Majorana fermions $\eta_1(x) = \gamma_x$ and $\eta_2(x) = \tilde{\gamma}_x$ for the convenience, $\epsilon^{\alpha,\beta}$ is rank-2 anti-symmetric tensor, $L$ is the size of the system, $k_F = \pi$ is the Fermi momentum. So this $L$-dependent long-range correlation originates from $k = 0$ zero modes at the Fermi surface. For free fermions, the correlation function encodes the information of entanglment spectrum [36,37]. It is reflected by a simple fact, $\langle \eta_\alpha(x)\eta_\beta(y) \rangle = \text{tr}(\eta_\alpha(x)\eta_\beta(y)\rho_O)$. Here $O$ is a subsystem and $x, y \in O$.

To find entanglement entropy, one has to find all eigenvalues of the correlation matrix, $\langle \eta_\alpha(x)\eta_\beta(y) \rangle$, with the subsystem of length $l$. Here from the long-range correlation in Eq. 10, we find that there is only one non-trivial eigenvalue that has the form $|1 - (2l/L)|$. One can obtain the corresponding value in the entanglement spectrum via solving $\tanh(\epsilon_0/2) = |1 - (2l/L)|$. This gives $\epsilon_0 = \log(L/l - 1)$. Subsequently, the $\epsilon_0$ results in the non-trivial entropy in Eq. 3. At the limit $l/L \ll 1$, the asymptotic expression of $S$ is $\sim l/L \log(l/L)$. The form is highly-nontrivial, as it is a non-logarithmic function. But its magnitude is weaker than a pure logrithmic function [38]. For detailed calculations, we refer the reader to Appendix B.

Zero-modes are present and influencing entanglement entropy in other contexts [39–41], including CFTs [42, 43]. But the effects of zero-modes are negligible in CFTs. On one hand, Eq. 10 is subleading relative to the $1/|x−y|$ decaying correlations in CFTs. On the other hand, the entanglement entropy in Eq. 3 is weaker than the logarithmic entropy. Thus the criticality $\Pi$ is a better platform to observe the effect of zero-modes in field theory rather than CFTs.

## 5 Velocity perturbation of the $z = 2$ Lifshtiz criticality

In this section, we exactly solve the noninteracting system described by the Hamiltonian Eq. 1 with open boundary conditions and in the presence of the velocity, $v$, perturbation to $u$. Here we derive the quantization condition corresponding to the system with open boundary conditions. This calculation supports the point made in the introduction suggesting that sgn($v$) can distinguish topological phases and also provides an asymptotic departure from the zero-velocity momentum quantization condition.

As the first step we consider the matrix equation for the eigenstates of the system

$$\begin{pmatrix} 0 & v\partial_x + u\partial_x^2 \\ -v\partial_x + u\partial_x^2 & 0 \end{pmatrix} \psi(x) = E\psi(x). \tag{11}$$

For a given energy $E = \epsilon_k = \sqrt{v^2k^2 + u^2k^4} \equiv uk\kappa$, where $\kappa = \sqrt{k^2 + v^2/u^2}$, there exists four normlaized eigenvectors corresponding to that energy level. These are given by

$$\psi_k(x) = \frac{1}{\sqrt{2L}} e^{ikx} \begin{pmatrix} \frac{-ivk+uk^2}{\epsilon_k} \\ -1 \end{pmatrix}, \quad \psi_{i\kappa}(x) = \sqrt{\frac{1}{1-e^{-2\kappa L}}} e^{-\kappa x} \begin{pmatrix} \sqrt{\kappa - v/u} \\ \sqrt{\kappa + v/u} \end{pmatrix}, \tag{12}$$

$$\psi_{-k}(x) = \frac{1}{\sqrt{2L}} e^{-ikx} \begin{pmatrix} \frac{ivk+uk^2}{\epsilon_k} \\ -1 \end{pmatrix}, \quad \psi_{-i\kappa}(x) = \sqrt{\frac{1}{e^{2\kappa L}-1}} e^{\kappa x} \begin{pmatrix} \sqrt{\kappa + v/u} \\ \sqrt{\kappa - v/u} \end{pmatrix}. \tag{13}$$

Under the open boundary conditions, we seek for the general solution of Eq. (11) in the form $\psi = C_1\psi_k + C_2\psi_{-k} + C_3\psi_{i\kappa} + C_4\psi_{-i\kappa}$, with real coeficients $C_i \in \mathbb{R}$, $i = 1\ldots 4$. These coefficients can be determined from the condition

$$\begin{pmatrix} e^{-i\theta} & e^{i\theta} & \sqrt{\kappa - v/u} & \sqrt{\kappa + v/u} \\ ike^{-i\theta} & -ike^{i\theta} & -\kappa\sqrt{\kappa - v/u} & \kappa\sqrt{\kappa + v/u} \\ -e^{ikL} & -e^{-ikL} & e^{-\kappa L}\sqrt{\kappa + v/u} & e^{\kappa L}\sqrt{\kappa - v/u} \\ -ike^{ikL} & ike^{-ikL} & -\kappa e^{-\kappa L}\sqrt{\kappa + v/u} & \kappa e^{\kappa L}\sqrt{\kappa - v/u} \end{pmatrix} \begin{pmatrix} C_1 \\ C_2 \\ C_3 \\ C_4 \end{pmatrix} = 0, \tag{14}$$

where $e^{-i\theta}$ is defined by $e^{-i\theta} = \frac{-iv/u+k}{\sqrt{v^2/u^2+k^2}}$. The existence of solution implies det$(M) = 0$, where $M$ is the $4 \times 4$ matrix in the left-hand-side of Eq. (14). This condition gives the quantization condition for the momenta in the presence of the finite velocity as

$$Q_\omega(y) \equiv \frac{2(1 + \frac{\omega^2}{y^2})}{\cosh\left(\sqrt{\omega^2 + y^2}\right)} + \left[\cos y \left(2 + 2\frac{\omega^2}{y^2} + \frac{\omega^4}{y^4}\right) - \left(2\frac{\omega}{y} + \frac{\omega^3}{y^3}\right)\sin y\right]$$

$$- \frac{\omega}{y}\sqrt{1 + \frac{\omega^2}{y^2}}\left[\left(2 + \frac{\omega^2}{y^2}\right)\cos y - \frac{\omega}{y}\sin y\right]\tanh\left(\sqrt{y^2 + \omega^2}\right) = 0, \tag{15}$$

where $y \equiv kL$ and $\omega \equiv Lv/u$. Analytically solving the transcendental Eq. 15 is a cumbersome task. The study of the properties of quantized $k$ is beyond the scope of the present paper, and we leave it for future works. However, one can perform an asymptotic analysis. Below we examine the asymptotic limits for both boundary and bulk modes.

- Consider the limit $|\omega| \gg 1$ and $y \ll 1$. This limit describes well the existence of boundary modes or absence thereof. In this limit, one finds that at positive $\omega > 0$, a real solution for $y$ exists and it is found as $y \simeq 2\omega e^{-\omega}$. For negative $\omega < 0$, there is no real solution for $y$, indicating the absence of boundary zero-mode. Further, one can estimate the energy of boundary modes at $\omega > 0$, which is given by

$$\epsilon_b = \frac{u}{L^2}\sqrt{\omega^2 y^2 + y^4} = \frac{u(2\omega e^{-\omega})}{L^2}\sqrt{\omega^2 + (2\omega e^{-\omega})^2} \simeq \frac{2u\omega^2 e^{-\omega}}{L^2}. \tag{16}$$

  If we express the energy in terms of the constants $u$ and $v$, we will have $\epsilon_b = 2u^{-1}v^2 e^{-Lv/u}$. The energy vanishes exponentially when $v/u \to \infty$. The exponential decay is the property corresponding to the topological degeneracy. We have also checked that the wavefunction is exponentially localized at the boundary. This concrete result supports our argument concerning the sign of $\mathrm{sgn}(v)$, discussed in the introduction.

- Consider the limit $1 < y \ll \omega$. In this limit, one finds that the quantization condition can be approximated by

$$Q_\omega(y_n + \delta y) \simeq (-1)^{n+1}\left(\frac{3\omega}{2y_n} - \frac{\omega^2}{2y_n^2}\delta y\right), \quad \text{where} \tag{17}$$

$$y_n = \frac{(2n+1)\pi}{2} \text{ is the CFT quantization condition}$$

  This indicates that the at large $\omega \gg 1$, the condition $Q_\omega(y_n + \delta y) = 0$ translates into $\delta y = 3y_n/\omega$. This, in turn, indicates that at large $\omega$, the $Q_\omega(y)$ converges to the CFT quantization condition at a "speed" of $3/\omega$ as $\omega \to \infty$.

These calculations represent the first step for deriving the finite-size scaling function describing deviation from the gapless Lifshitz point. However, the transcendent nature of the QC Eq. (15) makes it challenging to derive the form of the finite-size scaling function analytically. For this reason, we defer this analysis to forthcoming studies.

## 6  Conclusion

For the criticality $\Pi$ with quadratic dispersion, $\epsilon \sim \pm k^2$, we find a universal finite-size amplitude $J$ as the coefficient in front of $L^{-2}$ term in the ground state energy of the system. The magnitude of $J$ is anomalously large as it is of the order of one. There exist rich phenomena in finite-size scaling functions around this criticality [10, 33–35]. For example, with Eq. 1 at $v \neq 0$, a universal finite-size scaling emerges as a function of the scale $Lv$, and the function has a peak at the topological side. In principle, the Lifshitz criticality can also be enriched by symmetries. In that case, the presence of boundary modes around the Fermi surface and in case of breaking of some discrete symmetries (including the chiral symmetry and the time-reversal symmetry), one expects the emergence of a non-monotonic universal function of some scaling variable. This is an interesting and open problem.

The entanglement of the ground state is also found to be non-trivial, carrying a non-logarithmic entropy. This originates from the zero modes at the Fermi surface. Compared to CFTs, zero modes play a more crucial role at the criticality $\Pi$. This offers an opportunity to observe the effects of zero modes in the fermionic field theory [39–41]. Similarly, one can also explore the behaviors of entanglement entropy and boundary entropy [33, 44–47] around $\Pi$.

Effects of interactions are not explored in the present work. The exciting question is establishing the interacting theory of the low energy sector of $\Pi$ criticality. This question is beyond the scope of the Luttinger liquid [48, 49], where mostly the linear dispersion is considered.

## Acknowledgements

We thank Jin Zhang for valuable discussions on the entanglement entropy. We also thank insightful discussions with William Witczak-Krempa on the bosonic $z = 2$ Lifshitz criticality.

**Funding information** The research was supported by startup funds from the University of Massachusetts, Amherst.

## A  Derivations of the finite-size amplitude

In this section, we provide a detailed calculation of the universal finite-size amplitude, $J$. We start from the G.S. energy, given by Eq. 8, and rewrite it as

$$E_{G.S.} = -\frac{1}{2L^2} \oint_C \frac{dz}{2\pi i} z^2 \partial_z \ln\Big(\frac{e^{iz} + e^{-iz}}{2} + \frac{2}{e^z + e^{-z}}\Big). \tag{18}$$

Now let us consider the upper part of the contour, $C^+ = (i\delta, +2\pi L + i\delta)$, in the complex space of the integration variable $z$. The contribution to the $E_{G.S.}$ that corresponds to this upper part of the contour can be decomposed into

$$E_+ = -\frac{1}{2L^2} \int_{i\delta}^{2\pi L + i\delta} \frac{dz}{2\pi i} z^2 \partial_z \Big\{ \ln e^{-iz} + \ln\Big(\frac{e^{2iz} + 1}{2} + \frac{2e^{iz}}{e^z + e^{-z}}\Big)\Big\}. \tag{19}$$

The first term in the integral gives $(2\pi)^2/3L$, which is exactly the bulk energy. Thus, the finite size effect comes from the second term. We single out the finite-size contribution and define the integral as

$$J_+ \equiv \int_{i\delta}^{4\pi L + i\delta} \frac{dz}{2\pi i} z^2 \partial_z \ln\Big(\frac{e^{2iz} + 1}{2} + \frac{2e^{iz}}{e^z + e^{-z}}\Big). \tag{20}$$

Since the integrand at the arc is exponentially small, one could deform the contour into the form shoun by the (red) dashed line in Fig.4. Also, we define a new variable $z = x\alpha$ with $\alpha = e^{i\pi/4}$ and rewrite $J_+$ as

$$J_+ = -\int_0^{+\infty} \frac{dx}{4\pi} x^2 \partial_x \ln\Big(\frac{e^{2ix\alpha} + 1}{2} + \frac{2}{e^{\sqrt{2}x} + e^{-i\sqrt{2}x}}\Big). \tag{21}$$

Here $\alpha = e^{i\pi/4}$. The "$-$" sign emerges due to the deformation of the contour. Upon rescaling $x$ by $x \to \sqrt{2}x$, one obtains

$$J_+ = -\int_0^{+\infty} \frac{dx}{8\pi} x^2 \partial_x \ln\Big(\frac{e^{(i-1)x} + 1}{2} + \frac{2}{e^x + e^{-ix}}\Big). \tag{22}$$

Note there are also finite-size contributions coming from the other three segments, namely $(-i\delta, +2\pi L - i\delta)$ and $(\pm i\delta, -2\pi L \pm i\delta)$. Repeating similar calculations, one obtains the amplitude, $A$, presented in the Eq. 9 of the main text.

Here we further simplify Eq. 9 to obtain an expression that is more convenient for integration. One can take the derivative, $\partial_x$, and obtain

$$J = -\mathrm{Re} \int_0^{+\infty} \frac{x^2 dx}{\pi} \frac{\frac{1}{2}(i-1)(e^{-x} + e^{ix}) - 2(e^x + e^{-ix})^{-1}(e^x - ie^{-ix})}{\cos x + \cosh x + 2}. \tag{23}$$

Then one can take the real part of the integral and arrive at the following expression

$$J = \int_0^{+\infty} \frac{x^2 dx}{2\pi} \frac{1}{\cos x + \cosh x + 2} \left( e^{-x} + \cos x + \sin x + 4\frac{1 + (\cos x - \sin x)e^{-x}}{1 + 2\cos x e^{-x} + e^{-2x}} \right). \quad (24)$$

One can numerically evaluate the convergent integral over the single real-variable, $x$, and obtain the numerical value of the amplitude, $J$, presented in the manuscript.

# B  Derivation of the correlation function and the entanglement entropy

## B.1  The correlation function

Without losing generality, we start from a rotated Hamiltonian given by $H = \int_0^L dx \Psi^\dagger(x)\partial_x^2 \sigma_z \Psi(x)$. For BDI symmetry class, $\Psi(x) = [\psi(x), \psi^\dagger(x)]$. Under the periodic boundary conditions, the eigenstates are simply free wavefunctions. They can be cast in the form

$$\varphi_k(x) = \frac{1}{\sqrt{L}}e^{ikx}(0,1). \quad (25)$$

Thus the quasi-particle operator is given by

$$\hat{\varphi}_k = \frac{1}{\sqrt{L}}\sum_x e^{ikx}\psi^\dagger(x) = \frac{1}{2\sqrt{L}}\sum_x e^{ikx}[\eta_1(x) - i\eta_2(x)]. \quad (26)$$

As usual, we decomposed the fermion operator into Majorana fermions $\psi(x) = \frac{1}{2}[\eta_1(x) + i\eta_2(x)]$. Thus one can find that the Majorana operators diagonalizing the system are given by

$$b_k^1 = \frac{1}{\sqrt{L}}\sum_x (\eta_1(x)\cos kx + \eta_2(x)\sin kx), \quad (27)$$

$$b_k^2 = \frac{1}{\sqrt{L}}\sum_x (\eta_1(x)\sin kx - \eta_2(x)\cos kx). \quad (28)$$

With the diagonalizing Majorana fermions, one can evaluate the correlation function of $\eta$-Majorana fermions,

$$\langle \eta_\alpha(x_i)\eta_\beta(x_j)\rangle = I + \frac{i}{L}\sum_m \begin{pmatrix} -\sin k_m(x_i - x_j) & -\cos k_m(x_i - x_j) \\ \cos k_m(x_i - x_j) & -\sin k_m(x_i - x_j) \end{pmatrix}, \quad (29)$$

where summation over $m \in \mathbb{Z}$ has to be performed. This can be done for diagonal and off-diagonal independently as follows:

- For the diagonal matrix element one obtains a Kronecker $\delta$ upon switching to integration:

$$\langle \eta_1(x_i)\eta_1(x_j)\rangle \simeq i\delta_{i,j} - \frac{i}{2\pi}\int_{-\pi}^{\pi} dk \sin k(x_i - x_j) \simeq i\delta_{i,j}. \quad (30)$$

- Then we compute the non-diagonal correlators:

$$\langle \eta_1(x_i)\eta_2(x_j)\rangle = -\frac{i}{2\pi}\int_{-\pi}^{\pi} dk \cos k(x_i - x_j) = -i\delta_{i,j}. \quad (31)$$

The conclusion is that

$$\langle \eta_1(x_i)\eta_1(x_j)\rangle \simeq i\delta_{i,j}, \quad \langle \eta_1(x_i)\eta_2(x_j)\rangle \simeq -i\delta_{i,j}. \tag{32}$$

The absence of the $\sim 1/x$ decaying correlation, as opposed to the Ising CFT, is because the degeneracy here is rather trivial. Namely, $\varphi_k$ and $\varphi_{-k}$ share the same vector part, namely $(0,1)$. Recall that the degeneracy of $k$ and $-k$ in CFT is non-trivial: the vector parts of $\varphi_k$ and $\varphi_{-k}$ are two different eigenstates of $\sigma_z$.

Up to now, the correlation looked like the massive ones. Now we uncover a straightforward fact that there exists zero-mode, defined by

$$\hat{\chi}_0 = \frac{1}{\sqrt{L}}\sum_x \psi(x) = \frac{1}{2\sqrt{L}}\sum_x [\eta_1(x) + i\eta_2(x)]. \tag{33}$$

This zero-mode is *not* considered in the previous ground state. In the BDI class, the ground state is double degenerate. We define the first ground state $|GS\rangle_0$ to be

$$\hat{\psi}_k|GS\rangle_0 = 0, \quad k \in (-\pi, \pi). \tag{34}$$

In fact, this ground state is rather trivial. It is simply the vacuum state of the field operators $\psi(x)$. From the redundancy description, this ground state is simply filling the negative band. Now we define the second ground state as

$$|GS\rangle_1 = \chi_0^\dagger|GS\rangle_0. \tag{35}$$

The previous calculation concerned only $|GS\rangle_0$. Now we perform the calculation for $|GS\rangle_1$. The calculation is similar to the previous case with only difference coming from the zero mode. Namely, take the compensation of the contribution when the zero-mode is not occupied. It gives the correlation function evaluated with $|GS\rangle_1$,

$$\langle \eta_1(x_i)\eta_2(x_j)\rangle_1 \simeq -i\delta_{i,j} + \frac{2i}{L}(-1)^{x_i-x_j}. \tag{36}$$

One may use the following $2 \times 2$ matrix as the building block for the $2N \times 2N$ matrix:

$$\langle \eta_\alpha(x_i)\eta_\beta(x_j)\rangle_1 = \begin{pmatrix} \delta_{i,j} & -i\delta_{i,j} + \frac{2i}{L}(-1)^{x_i-x_j} \\ i\delta_{i,j} - \frac{2i}{L}(-1)^{x_i-x_j} & \delta_{i,j} \end{pmatrix}. \tag{37}$$

In the asymptotic limit $|x_i - x_j| \gg 1$, one obtains Eq. 10 of the main text.

## B.2 The entanglement entropy

We obtain the entanglement entropy for the free fermion system under consideration by directly diagonalizing the non-diagonal part of the correlation matrix. To this end, we consider the correlation matrix of the form

$$C_{i,j}^{\alpha,\beta} \equiv \begin{pmatrix} 0 & \delta_{ij} - \frac{2}{L}(-1)^{x_i-x_j} \\ -\delta_{ij} + \frac{2}{L}(-1)^{x_i-x_j} & 0 \end{pmatrix}. \tag{38}$$

The matrix is antisymmetric. Now let us reconstruct a single-particle Majorana fermion Hamiltonian that gives the correlation matrix Eq. (38):

$$H_C = \frac{i}{4}\sum_{i,j}\left(\delta_{ij} - \frac{2}{L}(-1)^{x_i-x_j}\right)\gamma_{1,i}\gamma_{2,j} + h.c.. \tag{39}$$

Here $\gamma_{\alpha,i}$ is an artificial Majorana fermion operator, and the index $\alpha = 1, 2$. One can prove that the eigenvalues of the $C$-matrix in Eq. (38) are the same as the ones for the $H_C$ operator. Then let us diagonalize the Hamiltonian by introducing the fermion operators, $a_i = (\gamma_{1,i} + i\gamma_{2,i})/2$. Such a substitution leads to the following quadratic fermionic Hamiltonian:

$$H_C = \sum_{j=1}^{l} a_j^\dagger a_j - \frac{2}{L} \sum_{i=1,j=1}^{l} (-1)^{x_i - x_j} a_i^\dagger a_j \,, \tag{40}$$

where $l$ is the size of subsystem we take. Here, let us focus on the second term and define

$$\hat{y} = \frac{1}{\sqrt{l}} \sum_{i=1}^{l} a_i e^{i\pi x_i} \,. \tag{41}$$

It is straightforward to check that $\hat{y}$ is still a fermionic operator, $\{\hat{y}, \hat{y}^\dagger\} = 1$. Thus $H_C$ becomes

$$H_C = \sum_{j=1}^{l} \left( a_j^\dagger a_j - \frac{1}{2} \right) - \frac{2l}{L} \left( \hat{y}^\dagger \hat{y} - \frac{1}{2} \right) \,. \tag{42}$$

This yielded the diagonal form of the fermionic Hamiltonian, $H_C$. The positive energies of the single particle spectrum thus read

$$\left| 1 - \frac{2l}{L} \right|, 1, 1 \ldots, 1 \,. \tag{43}$$

Note that $1 - \frac{2l}{L} \geq 0 \iff 2l \leq L$. If $2l > L$, the positive eigenvalue becomes $\frac{2l}{L} - 1$. Thus one can cast the non-trivial eigenvalues of $H_C$ in the form

$$1 - \frac{2l}{L}, \quad \text{if } 2l < L \,, \quad \frac{2l}{L} - 1, \quad \text{if } 2l > L \,. \tag{44}$$

This indicates that only zero modes introduce nontrivial correlation while all other modes give trivial correlation. If $\epsilon$ denotes the entanglement spectrum, we can find it from the solution of the following relation:

$$\tanh \frac{\epsilon}{2} = 1 - \frac{2l}{L} \,. \tag{45}$$

Thus the eigenvalues of the correlation matrix in a subsystem $l$ are the same with the number of 1's in the spectrum being $l-1$. Then we could find the entanglement spectrum from the set of values

$$\epsilon = \log \left( \frac{L}{l} - 1 \right), \quad \text{if } l \leq L/2 \,; \qquad \epsilon = \log \left( \frac{L}{L-l} - 1 \right), \quad \text{if } l \geq L/2 \,. \tag{46}$$

The entanglement entropy can be expressed as the sum over the entanglement spectrum [50, 51],

$$S = \sum_{n} \log \left( 2 \cosh \frac{\epsilon_n}{2} \right) - \frac{\epsilon_n}{2} \tanh \frac{\epsilon_n}{2} \,. \tag{47}$$

Here $\{\epsilon_n\}$ denotes the entanglement spectrum and the summation is performed over all values in the entanglement spectrum. Trivial terms corresponding to $\epsilon = \infty$ give zero contribution to the entropy. Thus we only need to keep track of the non-trivial contribution. Since there

is a symmetry between $l$ and $L-l$, we can only focus on the calculation for the case $2l < L$. Before performing the calculations, we need the following identities for any real variable $x$:

$$2\cosh\log x = x + \frac{1}{x}, \quad \tanh\log x = \frac{x^2-1}{x^2+1}. \tag{48}$$

Having these, one can take $x = \sqrt{L/l - 1}$ and putting together Eqs. 46, 47, 48, one finds

$$S = \log\left(\frac{L}{L-l}\right) + \frac{l}{L}\log\left(\frac{L}{l}-1\right). \tag{49}$$

This is the main result in the paper. Now we simplify Eq. 49 in two limiting cases:

- The limit of a small subsystem, $l \ll L$. Here first term in the expression for the entanglement entropy is approaching to zero due to the linear in $l/L$ prefactor. Namely,

$$\log\left(\frac{L}{L-l}\right) \simeq \frac{l}{L} + O(l^2/L^2), \quad \text{if } l \ll L. \tag{50}$$

Similarly, we have

$$\frac{l}{L}\log\left(\frac{L}{l}-1\right) \simeq \frac{l}{L}\log\frac{L}{l} + O(l^2/L^2), \quad \text{if } l \ll L. \tag{51}$$

Thus in this limit, within the $O(l/L)^2$ accuracy,

$$S \simeq \frac{l}{L}\ln\left(\frac{eL}{l}\right), \tag{52}$$

where $e$ is the Euler number. The asymptotic behavior is obviously non-logarithmic in $l$.

- The limit of a large subsystem, $2l \simeq L$. Firstly, exactly at $L = 2l$, the entanglement entropy is found to be $S = \log 2$. It is the typical result of a gapped system. Writing $\delta l = L/2 - l$, one thus has

$$\begin{aligned} S &= \log\left(\frac{1}{1/2+\delta l/2L}\right) + (1/2 - \delta l/L)\log\left(\frac{2}{1-2\delta l/L}-1\right) \\ &\simeq \log 2 + O\left(\frac{\delta l}{L}\right)^2. \end{aligned} \tag{53}$$

This indicates that the entanglement entropy, $S$, in this limit approaches its maximal value of $\log 2$ in this limit.

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
