# Peer review of "Universal finite-size amplitude and anomalous entangment entropy of $z=2$ quantum Lifshitz criticalities in topological chains"

_SciPost Physics, doi:SciPost Phys. 12, 134 (2022)_

## Round 1 · Referee Report · Anonymous (Referee 1) · 2021-12-8

Strengths
Interesting and universal results concerning non-conformal invariant criticalities.
Weaknesses
Lack of details and explanations about the derivation of the results.
Report
This paper deals with theories that at criticality are not conformal invariant, but they have a dynamical exponent $z=2$, rather than $z=1$ as occurs in conformal field theories (CFTs). Starting from these non-CFT criticalities, the authors try to investigate some universal features which can emerge in such one-dimensional systems. In particular, they focus on the finite-size corrections to the energy and on the entanglement entropy of the ground state. For the energy, they find a universal correction $\sim L^{-2}$, with $L$ the system size, while the entropy exhibits a non-logarithmic behaviour, $l/L\log(l/L)$, with $l$ the subsystem size, due to the presence of zero-modes at the Fermi surface of the considered models.
They benchmark their results against lattice computations which involve a generalisation of the Majorana chain and of the Su-Schrieffer–Heeger (SSH) model.
The paper is well-written and it contains some nontrivial results. Therefore, I would recommend it for publication once a minor revision work has been done. Indeed, I would suggest to add more details and explanations such that the work can be self-consistent and more pedagogical.
Here is a short list of comments/questions/typos:
- Typo in the title: entangment $\rightarrow$ entanglement;
- Pag. 4 before Eq. (5) and after Eq. (6): Hamilotnian $\rightarrow$ Hamiltonian (and hamiltonian $\rightarrow$ Hamiltonian pag. 2);
- Pag. 5: "the computation of the finite-size amplitude of the ground state is similar to the method used in references [10,32]": could the author explain in more details the numerical method used to obtain the universal coefficient $A$?
- Pag. 6 after Eq. (6): "a is the lattice space": where does $a$ enter in the text?
- Pag. 6: "entanglment" $\rightarrow$ entanglement;
- Pag. 7: it would be more clear if you could comment the steps from the correlation function to the non-trivial value of the entanglement espectrum, $\epsilon_0$;
- "Zero-modes are present and influencing entanglement entropy in other contexts, including CFTs": the authors could be interested into another scenario where the presence of a zero mode at the conformal point of a free scalar theory affects the behaviour of the entanglement entropy (J.Stat.Mech.0512:P12012,2005).\\
- Do the authors have any insights about what happens in higher dimensional systems? For example, free massless non-relativistic fermions show logarithmic violations of the area law.
Here we respond to comments made by the Referee in the same order they appear in the report.
-
We thank the Referee for the detailed description of the work, finding that the paper is well-written, and suggesting the publication.
-
We appreciate the Referee's helpful suggestions. The manuscript has been updated with further information. Now it includes more explanations and two new appendices with details of calculations.
-
Typos in the title and around Eqs. 5, 6 are corrected. Some other typos are also corrected.
-
We discuss the details of the numerical method to estimate the finite-size effects in the present version (particularly in Sect 3). We also added Ref. 36, discussing some specific details.
-
Again, we thank the Referee for pointing to these and other typos. The sentence "$a$ is the lattice space" is deleted (which was a leftover from an older version). We have corrected this and many other typos present in the text.
-
We have added the details about the steps discussing the calculation of the entanglement spectrum from the correlation function.
-
The information on the paper J.Stat.Mech.0512:P12012,2005 is valuable. We have introduced the corresponding discussion and included the important reference.
-
We thank the Referee for pointing out this interesting question of higher dimension. Although the generalization to higher dimensions is of great interest, we at this moment do not have a good understanding of such generalization. For that reason, we prefer not to speculate about it in the present paper.
Bests The authors
Author: Ke Wang on 2022-03-18 [id 2298]
(in reply to Report 2 on 2022-02-08)Here we respond to comments made by the Referee in the same order they appear in the report.
We thank the Referee for the concise description of the work and for suggesting the publication.
The suggestion of exposing more details is valuable. We have introduced several detailed explanations and calculations into the present version of the manuscript. The summary of changes is contained in the List of Changes.
Also, the suggestion about exploring the deviations from the gapless point is important. We have added a new section (Sect. 5 titled Velocity perturbation of the $z = 2$ Lifshitz criticality) to the manuscript. This section contains a discussion of the deviation from the Lifshitz point.
Bests The authors

---

## Round 1 · Referee Report · Anonymous (Referee 2) · 2022-2-8

Report
The paper addresses finite size scaling at a topological phase transition with a degenerate dispersion relation \epsilon \sim k^2. There is a body of well established literature, starting from the universal finite size scaling at a conformal critical point, where the correction is known to be proportional to the central charge of the theory. These finding were subsequently generalized to an entire universal scaling function, covering small deviation from the conformal critical point. All these findings, however, are restricted to the linear Dirac dispersion relation. While most prominent, the Dirac dispersion is not unique. There are instances (e.g. multicrtical points) where the low-energy dispersion relation degenerates into k^2. The present paper presents results for finite size scaling of grounsntate energy and entanglement entropy in such situation. This
is a welcome contribution to the field and a valuable addition to the existing body of knowledge. I support its publication.
As a suggestion (in agreement with the first referee): the manuscript can benefit from language proof-reading and exposing more details of the calculations. Authors thought about deriving scaling function, covering deviations from the gapless point would be a welcome addition too.

---

## Round 2 · Author Response

Dear Editor

Thank you for forwarding us the revision request on Feb. 8, 2022. We are grateful to both Referees for constructive reports and strongly supporting publication. We have carefully considered all the comments and questions raised by Referees.

Our detailed response to the comments and questions raised by the Referees is presented below. The Referee recommendations are primarily targeting the improvement of the presentation and the inclusion of more technical details. We address this point in the present version of the manuscript, where more supporting and technical materials are added. The additions include more detailed explanations in the main part of the manuscript, one additional section on perturbations around the criticality, and two new appendices.

Having met Referee recommendations, we kindly request further consideration of publication in the SciPost Physics.

Sincerely,

The authors.

Response to Referee. II

  1. " The paper addresses finite size scaling at a topological phase transition with a degenerate dispersion relation $\epsilon \sim k^2.$ There is a body of well established literature, starting from the universal finite size scaling at a conformal critical point, where the correction is known to be proportional to the central charge of the theory. These finding were subsequently generalized to an entire universal scaling function, covering small deviation from the conformal critical point. All these findings, however, are restricted to the linear Dirac dispersion relation. While most prominent, the Dirac dispersion is not unique. There are instances (e.g. multicrtical points) where the low-energy dispersion relation degenerates into $k^2$. The present paper presents results for finite size scaling of ground state energy and entanglement entropy in such situation. This is a welcome contribution to the field and a valuable addition to the existing body of knowledge. I support its publication."

Response: We thank the referee for the concise description of the work and suggesting the publication.

  1. "As a suggestion (in agreement with the first referee): the manuscript can benefit from language proof-reading and exposing more details of the calculations."

Response: We have introduced several detailed explanations and calculations into the present version of the manuscript. See List of changes.

  1. "Authors thought about deriving scaling function, covering deviations from the gapless point would be a welcome addition too"

Response: We thank the reviewer for this suggestion. A new section (Sect. number 5 entitled Velocity perturbation of the $z = 2$ Lifshitz criticality) is added to the manuscript. This section contains the discission of the deviation from the Lifshitz point.

Response to Referee I. 1. " This paper deals with theories that at criticality are not conformal invariant, but they have a dynamical exponent $z=2$ , rather than $z=1$ as occurs in conformal field theories (CFTs). Starting from these non-CFT criticalities, the authors try to investigate some universal features which can emerge in such one-dimensional systems. In particular, they focus on the finite-size corrections to the energy and on the entanglement entropy of the ground state. For the energy, they find a universal correction $\sim 1/L^2$ and $L$ is the system size, while the entropy exhibits a non-logarithmic behaviour, due to the presence of zero-modes at the Fermi surface of the considered models. They benchmark their results against lattice computations which involve a generalisation of the Majorana chain and of the Su-Schrieffer–Heeger (SSH) model. The paper is well-written and it contains some nontrivial results. Therefore, I would recommend it for publication once a minor revision work has been done. "

Response: We thank the referee for the detailed description of the work, finding that the paper is well-written, and suggesting the publication.

2." Indeed, I would suggest to add more details and explanations such that the work can be self-consistent and more pedagogical. "

Response: We appreciate the Referee's helpful suggestions. The manuscript has been updated with further information. Now it includes more explanations as well as two new appendices with details of calculations.

  1. "- Typo in the title: entangment $\rightarrow$ entanglement; Pag. 4 before Eq. (5) and after Eq. (6): Hamilotnian $\rightarrow$ Hamiltonian (and hamiltonian $\rightarrow$ Hamiltonian pag. 2); "

Response: Typos are corrected.

    • Pag. 5: "the computation of the finite-size amplitude of the ground state is similar to the method used in references [10,32]": could the author explain in more details the numerical method used to obtain the universal coefficient A ?

Response: We discuss the details of the numerical method to estimate the finite-size effects in the present version (particularly in Sect 3). We also added Ref. 36 discussing some specific details.

    • Pag. 6 after Eq. (6): "$a$ is the lattice space": where does $a$ enter in the text?
  1. Pag. 6: "entanglment" $\rightarrow$ entanglement; }

Response: Again we thank the Referee for pointing to these and other typos. The sentence "$a$ is the lattice space" is deleted (which was a leftover from an older version). We have corrected this and many other typos present in the text.

  1. "Pag. 7: it would be more clear if you could comment the steps from the correlation function to the non-trivial value of the entanglement espectrum"

Response: We have added the details about the steps discussing the calculation of the entanglement spectrum from the correlation function.

  1. "Zero-modes are present and influencing entanglement entropy in other contexts, including CFTs": the authors could be interested into another scenario where the presence of a zero mode at the conformal point of a free scalar theory affects the behaviour of the entanglement entropy (J.Stat.Mech.0512:P12012,2005)."

Response: We thank the Referee for the valuable information. We have introduced the corresponding discussion and included the important reference.

  1. " Do the authors have any insights about what happens in higher dimensional systems? For example, free massless non-relativistic fermions show logarithmic violations of the area law."

Response: We thank the Referee for pointing to this interesting question. However unfortunately, although the generalization to higher dimensions is of great interest, we at this moment do not have a good understanding of such generalization. For that reason, we prefer not to speculate about it in the present paper.

---

## Round 2 · List of Changes

1. A comment on the derivation of the boundary condition is added above Eq.~7 in the manuscript.

2. A detailed description of the passage from the correlation function to the entanglement spectrum is added below Eq.~10.

3. A new section titled " 5: Velocity perturbation of the $z = 2$ Lifshitz criticality" is added.

4. A new Appendix titled "A Derivations of the finite-size amplitude" is added.

5. A new Appendix titled "Appendix B Derivations of the correlation function and the entanglement entropy" is added.

---

## Editorial Decision

published